


# 1 Brief Communication: A case study of risk assessment for
# 2 facilities associated with earthquake-induced liquefaction
# 3 potential in Kimhae City, South Korea

**Sang-Soo Jeon[1], Daeyang Heo[2], Sang-Seung Lee[3]**
[1] Department of Civil & Urban Engineering, Construction Technology Research Center, INJE
University, Inje-ro 197, Kimhae City, Gyeongsangnam-do, 50834, South Korea
[2]Industrial Site Division, Gyeongsangnam-do Provincial Government, 200 Jungangdae-ro, Uichang-gu,
Changwon City, Gyeongsangnam-do, 51154, South Korea
[3]Kyong-Ho Engineering, Kyongho Building, 41 Cheyukgwan-ro 74 Beon-gil, Guri, Gyeonggi-Do,
11940, South Korea
*Correspondence to:* Sang-Soo Jeon (ssj@inje.ac.kr)

**Abstract** Liquefaction causes secondary damage after earthquakes; however, liquefaction related phenomena were
rarely reported until after the $M_w = 5.4$ November 15, 2017 Pohang earthquake in Korea. Both the $M_w = 5.8$ September
12, 2016 Gyeongju earthquake and $M_w = 5.4$ November 15, 2017 Pohang earthquake occurred in the fault zone of
Yangsan City (located in the south-eastern part of South Korea), and both of these earthquakes induced liquefaction.
Moreover, they demonstrated that Korea is not safe against the liquefaction induced by earthquakes. In this study,
estimations and calculations were performed based on the distances between the centroids of administrative districts
and an epicenter located at the Yangsan Fault, the peak ground accelerations (PGAs) induced by $M_w = 5.0$ and 6.5
earthquakes, and a liquefaction potential index (LPI) calculated based on groundwater level and standard penetration
test results from 274 locations in Kimhae City (adjacent to the Nakdong river and across the Yangsan Fault). Then, a
kriging method using geographical information systems was used to evaluate the liquefaction effects on the risk levels
of facilities. The results indicate that a $M_w = 5.0$ earthquake induces a small and low level of liquefaction, resulting in
slight risk for facilities, but a $M_w = 6.5$ earthquake induces a large and high level of liquefaction, resulting in a severe
risk for facilities.

## 1  Introduction
Soil liquefaction occurs when the strength of soils (in areas with a high level of groundwater and loose sand or sandy
soils) is reduced by applied earthquake loading. A loss of shear strength occurs because the effective stress is reduced
as excess pore water pressure is increased and gradually decreased when earthquake loading is applied (Kramer,
1996; Youd and Idriss, 2001).
The soil liquefaction induced by the Pohang earthquake was reported as a first case in Korea; however, liquefaction
has occurred following various earthquakes, including the Niigata earthquake ($M_w = 7.6$) in 1964, Loma Prieta
earthquake ($M_w = 6.9$) in 1989, Northridge earthquake ($M_w = 6.7$) in 1994, Tohoku earthquake ($M_w = 9.1$) in 2011,
and Christchurch earthquakes ($M_w = 6.2$–7.1) in 2010 and 2011. Earthquakes resulted in substantial amounts of
infrastructure damage, such as building damage induced by differential settlements, the lateral displacement of roads,
and lifeline damage. The structural and foundation performances of facilities subjected to settlement and tilt when
subsurface layers of soils are liquefiable have been analyzed to estimate the resulting damage (Bakir and Karasin,
2016; Bray and Dashti, 2010; Bullock et al., 2019; Hayden, 2014; Kamao et al., 2014; Lanzano et al., 2014; Lu et
al., 2017; Wakamatsu and Numata, 2004; Zupan, 2014). Other studies have constructed soil liquefaction hazard maps
to determine land damage and/or analyze liquefaction potential (Ballegooy et al., 2012; Habibullah et al., 2012; Naik
et al., 2020; Ziabari et al., 2017).
A liquefaction potential index (LPI) has also been used to estimate the risk levels of facilities with respect to
liquefaction (Holzer, 2008; Iwasaki et al., 1982). The LPI is based on a factor of safety (FS) calculated based on the



groundwater level and peak ground acceleration (PGA) induced by earthquake loading, and it represents the
liquefaction potential. There is no liquefaction when the FS is equal to or greater than 1.0; by contrast, it has the
potential for liquefaction when the FS is less than 1.0. However, a liquefaction potential estimated using the FS
cannot represent the ground damage for broad areas; rather, it is only applicable to local specific areas. The LPI
proposed by Iwaski et al. (1982) has been used to estimate the hazards induced by liquefaction in broad areas and to
produce corresponding hazard maps (Chung and Rogers, 2011; Iwasaki et al., 1982; Lee et al., 2003).
When an earthquake occurs, the liquefaction potential is determined by the groundwater level and PGA associated
with the ground characteristics. In this study, the safety of facilities in Kimhae City (located in the south-eastern part
of Korea) was estimated based on attenuation equations associated with the distance from the epicenter to the centroid
of seventeen administrative districts in Kimhae City. The Pohang earthquake, the largest recent earthquake in Korea,
had a magnitude of 5.0. An earthquake magnitude of 6.5, corresponding to a PGA of 0.2g, is the standard for the
design of earthquake-resistant structures in Korea. Therefore, in this study, the FS values for facilities in Kimhae
City were estimated for $M_w$ 5.0- and 6.5-earthquakes, and the liquefaction potential was evaluated based on currently
available standard penetration test (SPT) results. Since cone penetration test (CPT) results can reflect more precise
ground conditions, in the future, liquefaction potential values should be revised based on CPT results to estimate the
risk levels of facilities. Moreover, attenuation relationships should be developed to reflect the widely distributed
transgressive sands in Kimhae City.
**2   Liquefaction Potential Index (LPI)**
In this study, the LPI proposed by Iwasaki et al. (1978) was used to estimate the ground damage level induced by
liquefaction. As described in Eqn. (1), the LPI is calculated based on the ground depth and characteristics of soil, as
follows:

$$\text{LPI} = \int_0^{20} F(z)W(z)dz \qquad (1)$$

In this equation, z represents the ground depth, and F(z) is a function of the FS for liquefaction. If FS ≤ 1.0, F(z)
= (1- FS), and if FS > 1.0, F(z) = 0. W(z) = (10 − 0.5 z) and W(z) = 0 for z ≤ 20 m and z > 20 m, respectively. Eqn.
(1) provides LPIs in the range from 0 to 100. Iwasaki et al. (1978) proposed levels of liquefaction severity, as
described in Table 1, associated with 63 and 22 areas at liquefaction and non-liquefaction sites, respectively.

Table 1. Level of liquefaction severity based on liquefaction potential index (LPI) (Iwasaki et al., 1982)

| LPI | Severity |
| --- | --- |
| 0 | Very low |
| 0 < LPI ≤ 5 | Low |
| 5 < LPI ≤ 15 | High |
| 15 < LPI | Very high |

The LPI is determined by integrating F(z) multiplied by W(z) from the ground surface to a ground depth of 20 m,
and a single value corresponding to a site is evaluated. The LPI can be evaluated for each layer of soil. For example,
if a non-liquefaction layer such as bed rock exists in the soil layers within 20 m of ground depth, the ground depth
for calculating the LPI is estimated from the ground surface to the depth susceptible to liquefaction.
A simplified method for estimating the FS of liquefaction was proposed by Seed and Idriss (1971), as follows:

$$\text{FS} = \frac{\text{CRR}}{\text{CSR}} \times \text{MSF} \qquad (2)$$

The cyclic resistance ratio (CRR) and cyclic stress ratio (CSR) represent the capacity of soil to resist liquefaction
and the ratio of the shear stress relative to the effective vertical overburden stress, respectively. The magnitude scaling
factor (MSF) varies with the magnitude of the earthquake. In this study, as shown in Figure 1, a flowchart is used to
determine the LPI values. The CSR and CRR are calculated based on the SPT results and soil parameters, respectively.



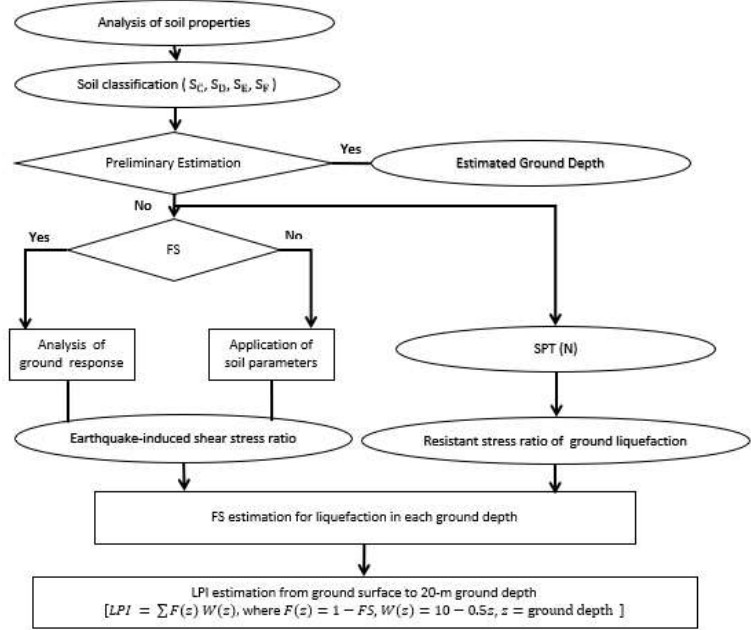


Figure 1. Flowchart for estimating liquefaction potential index (LPI) (Choe and Ku, 2009)
## 3 Estimation of peak ground acceleration (PGA)
The PGA induced by an earthquake has large variations associated with the soil characteristics, distance from the
epicenter, and ground depth. As the PGA is a crucial factor, it is directly used to evaluate earthquake-induced damage.
The largest PGA normally occurs near the epicenter, and the PGA generally decreases as the distance from the
epicenter increases. In this study, the PGA was evaluated based on both the distance from each administrative district
to the epicenter and an attenuation relationship; then, the risk levels of facilities affected by earthquake-induced
liquefaction were evaluated.
### 3.1 Estimation of the location of epicenter and distance from epicenter to each administrative district
Figure 2 shows Kimhae City with respect to the active Yangsan Fault. As shown in Figure 2(a), the fault lies across
the study area (Kimhae City), and the horizontally extended location from the centroid of Kimhae City to the closest
fault is assumed to be the location of the epicenter. The distance from the centroid of Kimhae City to the epicenter
is 16.8 km. There are seventeen administrative districts in Kimhae City. The distances from the epicenter to the
centroid of each administrative district were calculated. Figure 2(b) shows an example of how the distance of 3.6 km
from Daedong-myun to the epicenter was calculated. Table 2 describes the distances from the centroid of each
administrative district to the epicenter.





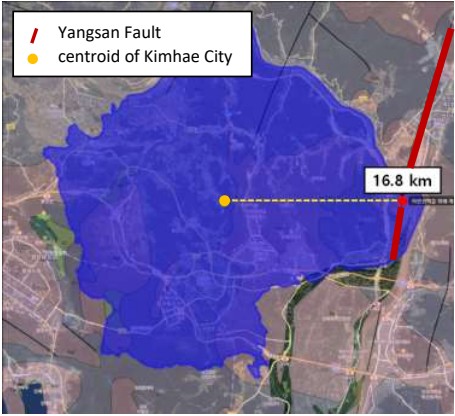

(a) Distance from epicenter to the centroid of Kimhae City

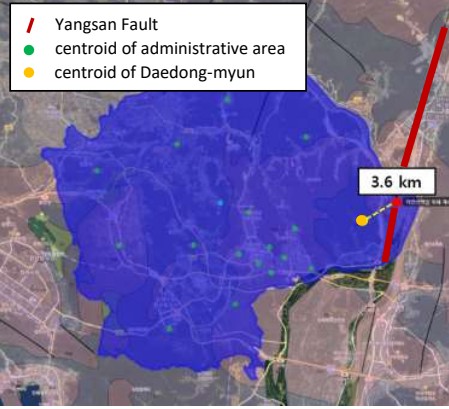

(b) Distance from epicenter to the centroid of Daedong-myun
Figure 2. Distance from epicenter to the centroid of Kimhae City and Daedong-myun, respectively.




Table 2. Distance from Yangsan Fault to centroid of each administrative district

| Administrative district | Distance from Yangsan fault (km) |
|---|---|
| Daedong-myeon | 3.6 |
| Saman-dong | 10.1 |
| Buram-dong | 10.3 |
| Sangdong-myeon | 10.6 |
| Hwalcheon-dong | 11.9 |
| Dongsang-dong | 12.8 |
| Buwon-dong | 13.8 |
| Bukbu-dong | 14.2 |
| Hoehyeon-dong | 14.5 |
| Chilsanseobu-dong | 18.1 |
| Naeoe-dong | 18.8 |
| Saengnim-myeon | 18.8 |
| Juchon-myeon | 19.8 |
| Hallim-myeon | 21.7 |
| Jangyu-myeon | 24.8 |
| Jillye-myeon | 27.0 |
| Jinyeong-eup | 28.7 |

### 3.2 Attenuation relationship of PGA

Three of the most reliable attenuation relationships for the PGA have been proposed for use by the Ministry of the Interior and Safety of Korea (Choi et al., 2005; Jo and Baag, 2003; Lee et al., 2003). The most reliable attenuation relationship proposed by Choi et al. (2005) was used in this study. The attenuation relationship proposed by Choi et al. (2005) is compared to those proposed by Midorikawa (2004) and Munson (1997) for an earthquake magnitude of 5.0; it is found that the PGAs obtained from the attenuation relationship proposed by Choi et al. (2005) are highly similar to those obtained from the relationship proposed by Midorikawa (2004), but different from those obtained from Munson (1997), with the latter being based on ground conditions in Hawaii. As the calculated values are shown in Figure 3, as there were no available data corresponding to a distance of less than 10 km and the attenuation relationship proposed by Choi et al. (2005) resulted in the overprediction of the PGAs. Therefore, the attenuation relationship was considered as unreliable within a 10-km distance from the epicenter. Eqn. (3) expresses the attenuation relationship proposed by Choi et al. (2005), and Table 3 describes the parameters of the attenuation relationship for estimating PGAs.

$$lnPGA\left(\frac{cm}{sec^2}\right) = c_0 + c_1R + c_2lnR - \ln[\min(R, 100)] - \frac{1}{2}\ln[\max(R, 100)] \tag{3}$$

In the above, R represents the distance from the epicenter, and $c_k(0,1,2) = \xi_0^k + \xi_1^k(M_w - 6) + \xi_2^k(M_w - 6)^2 + \xi_3^k(M_w - 6)^3$ for k = 0, 1, and 2.



Table 3. Parameters of the attenuation relationship for estimating PGA (Jo and Baag, 2003)

| | $\xi_0^0$ | $\xi_0^1$ | $\xi_0^2$ | $\xi_1^0$ | $\xi_1^1$ | $\xi_1^2$ | $\xi_2^0$ | $\xi_2^1$ | $\xi_2^2$ | $\xi_3^0$ | $\xi_3^1$ | $\xi_3^2$ |
|---|---|---|---|---|---|---|---|---|---|---|---|---|
| PGA | 0.1073829 E+02 | -0.2379955 E-02 | -0.2437218 E+00 | 0.5909022 E+00 | 0.2081359 E-03 | 0.9498274 E-01 | -0.5622945 E-01 | -0.2046806 E-04 | -0.8804236 E-02 | 0.2135007 E-01 | 0.4192630 E-04 | -0.3302350 E-02 |


The SPT data of 903 locations, provided by both the geotechnical information database system of a governmental
organization and construction companies, were collected to estimate the LPI values in the study area. Since some of
the important SPT data were missing, a reliable dataset of 274 locations was selected, and then a geographical
information system was used to plot the locations of the selected SPT data. The locations of SPT linearly arrayed
inside of the dotted line may result in the deviation of contour lines of LPI as shown in Figure 4. The SPT data
recorded at the various coordinates and the kriging method were used to construct the contour lines of the LPI values.

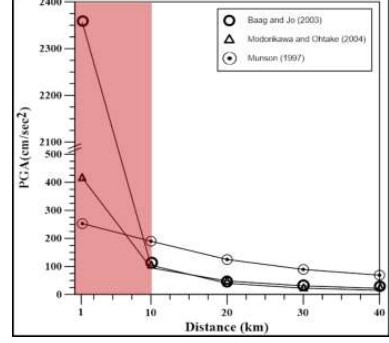

Fig. 3. Peak ground acceleration (PGA) vs. distance from epicenter      Fig. 4. Location of standard penetration test
(SPT) used to estimate LPI


**4   Risk level of facilities in Kimhae City**
Facilities in Kimhae City are categorized as described in Table 4.
Table 4. Facilities in Kimhae City

| Facility | Number or length |
|---|---|
| Tunnel | 15 |
| Bridge | 412 |
| Light rail transit (km) | 24.6km |
| Railway (km) | 91.3km |
| Road (km) | 1,145.3km |
| Water pipe (km) | 1,340.0km |
| Sewage pipe (km) | 1,502.0km |
| Public facility | 96,729 |
| Shelter outside a building | 27 |


### 4.1 Spatial distribution of LPI for $M_w$ = 5.0 and 6.5 earthquakes
Figures 5(a) and (b) show the LPI distribution and Figures 5(c) and (d) show the ratio of the covered area with
respect to the range of the LPI values for $M_w$ = 5.0 and 6.5 earthquakes, respectively.
The "very high" and "high" level of liquefaction severity for the $M_w$ = 5.0 earthquake cover 2 km$^2$ (0.2%) and 22.1
km$^2$ (4.8%) of the study area, respectively. The "very high" and "high" level of liquefaction severity for the $M_w$ = 6.5
earthquake cover 28.6 km$^2$ (6.2%) and 11.5 km$^2$ (2.5%) of the study area, respectively. These areas seem to be small
in proportion to the total area, but are not small in proportion to the plat area. As the earthquake magnitude increases
from $M_w$ = 5.0 to $M_w$ = 6.5, the proportion of land with high level of liquefaction severity increases substantially.
Figure 6 shows bridges, buildings, and water pipelines superimposed on the spatial distribution of the LPI for both
the $M_w$ = 5.0 and 6.5 earthquakes. Figure 7 shows how facilities are distributed in level of liquefaction severity zones.
As we expected, much greater proportions of facilities are distributed in high level of liquefaction severity areas for
the $M_w$ = 6.5 earthquake relative to those for the $M_w$ = 5.0 earthquake.

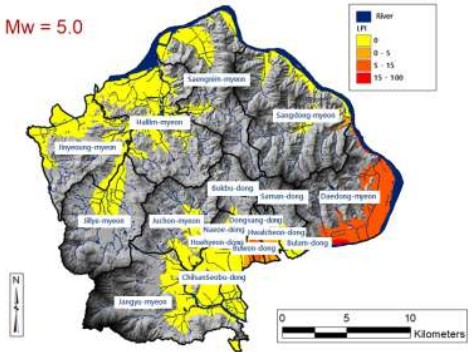
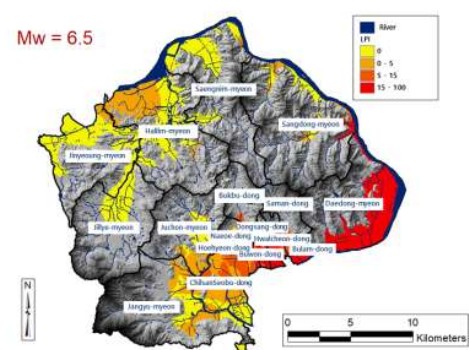
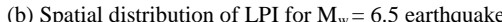

(a) Spatial distribution of LPI for $M_w$ = 5.0 earthquake    (b) Spatial distribution of LPI for $M_w$ = 6.5 earthquake

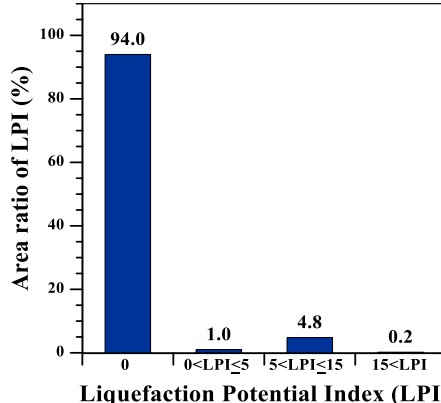
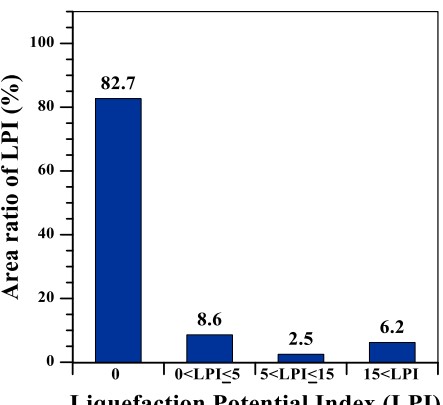

(c) Area ratio of LPI for $M_w$ = 5.0 earthquake    (d) Area ratio of LPI for $M_w$ = 6.5 earthquake

Figure 5. Spatial distribution and area ratio of LPI for $M_w$ = 5.0 and 6.5 earthquakes, respectively




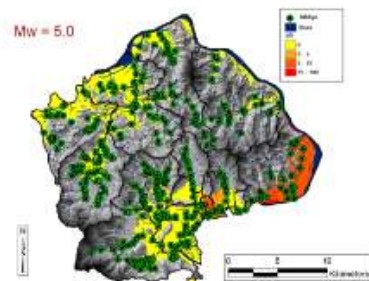

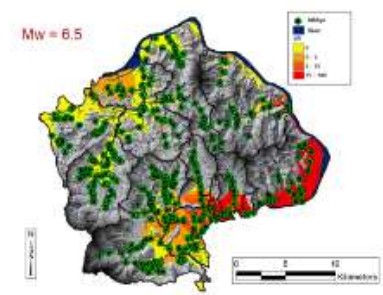

(a) Bridges superimposed on spatial distribution
of LPI for $M_w = 5.0$ earthquake

(b) Bridges superimposed on spatial distribution
of LPI for $M_w = 6.5$ earthquake

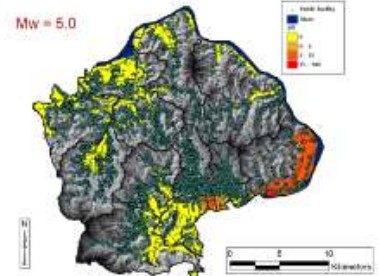

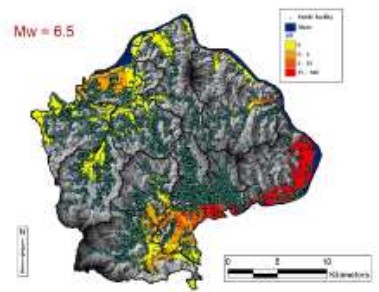

(c) Public facilities superimposed on spatial
distribution of LPI for $M_w = 5.0$ earthquake

(d) Public facilities superimposed on spatial
distribution of LPI for $M_w = 6.5$ earthquake

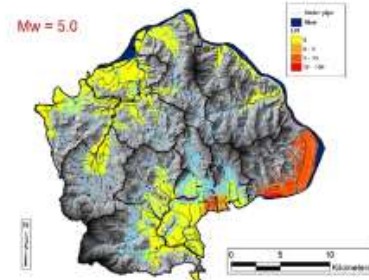

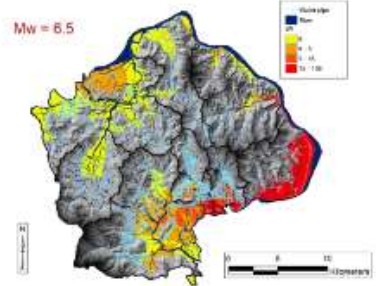

(e) Water pipelines superimposed on spatial
distribution of LPI for $M_w = 5.0$ earthquake

(f) Water pipelines superimposed on spatial
distribution of LPI for $M_w = 6.5$ earthquake


Figure 6. Bridges, buildings, and water pipelines superimposed on spatial distribution of LPI for $M_w = 5.0$ and
6.5 earthquakes, respectively





(a) Bridges with respect to LPI for $M_w$ = 5.0 earthquake

(b) Bridges with respect to LPI for $M_w$ = 6.5 earthquake

(c) Public facilities with respect to LPI for $M_w$ = 5.0 earthquake

(d) Public facilities with respect to LPI for $M_w$ = 6.5 earthquake

(e) Water pipelines with respect to LPI for $M_w$ = 5.0 earthquake

(f) Water pipelines with respect to LPI for $M_w$ = 6.5 earthquake

Figure 7. Bridges, buildings, and water pipeline with respect to LPI for $M_w$ = 5.0 and 6.5 earthquakes.





**4.2 Risk assessment of facilities with respect to LPI for $M_w = 5.0$ and $M_w = 6.5$ earthquakes**

In general, most facilities are distributed where the LPI = 0. For example, 11.2% of light rail transit facilities and 5.0% of sewage pipelines are distributed in areas with low level of liquefaction severity. Moreover, 7.0% of bridges, 9.2% of light rail transit facilities, 5.4% of roadways, and 6.2% of buildings are distributed in areas with high level of liquefaction severity, whereas only 0.1% of roadways, sewage pipelines, and buildings are distributed in areas with very high level of liquefaction severity. Table 5 shows the ratios of facilities corresponding to various LPI ranges for the $M_w = 5.0$ earthquake. As the earthquake magnitude increases from 5.0 to 6.5, the risk levels of facilities increase. Notably, 93.3% of tunnels, 25.7% of light weight transit facilities, and 6.7% to 31.2% of other facilities are in areas with very low level of liquefaction severity. The facilities with both low and very high level of liquefaction severity comprise approximately 10% of the study area. The length of light weight transit in areas with very high level of liquefaction severity is approximately 7.0 km (28.6%), and is longer than 6.3 km (25.7%) in areas with very low level of liquefaction severity. Table 6 shows the ratios of facilities corresponding to various level of liquefaction severity ranges for the $M_w = 6.5$ earthquake.

Table 5. Ratios of facilities covered by LPI for $M_w = 5.0$ earthquake

| Facility \ LPI | 0 | 0-5 | 5-15 | 15-100 |
|---|---|---|---|---|
| Tunnel, number (%) | 15 (100) | 0 (0.0) | 0 (0.0) | 0 (0.0) |
| Bridge, number (%) | 369 (89.6) | 14 (3.4) | 29 (7.0) | 0 (0.0) |
| Light rail transit, km (%) | 19.6 (79.6) | 2.8 (11.2) | 2.2 (9.2) | 0.0 (0.0) |
| Railway, km (%) | 91.3 (100.0) | 0.0 (0.0) | 0.0 (0.0) | 0.0 (0.0) |
| Road, km (%) | 1,041.2 (90.9) | 41.2 (3.6) | 61.8 (5.4) | 1.1 (0.1) |
| Water pipeline, km (%) | 1,181.9 (88.2) | 48.2 (3.6) | 109.9 (8.2) | 0.0 (0.0) |
| Sewage pipeline, km (%) | 1,357.8 (90.4) | 75.1 (5.0) | 67.6 (4.5) | 1.5 (0.1) |
| Public facility, number (%) | 86,862 (89.8) | 3,772 (3.9) | 5,997 (6.2) | 98 (0.1) |
| Shelter outside a building, number (%) | 24 (88.9) | 1 (3.7) | 2 (7.4) | 0 (0.0) |

Table 6. Ratios of facilities covered by LPI for $M_w = 6.5$ earthquake

| Facility \ LPI | 0 | 0-5 | 5-15 | 15-100 |
|---|---|---|---|---|
| Tunnel, number (%) | 14 (93.3) | 0 (0.0) | 1 (6.7) | 0 (0.0) |
| Bridge, number (%) | 278 (67.5) | 68 (16.5) | 25 (6.1) | 41 (9.9) |
| Light rail transit, km (%) | 6.3 (25.7) | 2.8 (11.5) | 8.5 (34.2) | 7.0 (28.6) |
| Railway, km (%) | 76.2 (83.5) | 14.5 (15.9) | 0.6 (0.6) | 0.0 (0.0) |
| Road, km (%) | 714.5 (62.4) | 189.5 (16.6) | 117.8 (10.3) | 123.5 (10.7) |
| Water pipeline, km (%) | 863.4 (64.4) | 188.0 (14.1) | 143.6 (10.7) | 145.0 (10.8) |
| Sewage pipeline, km (%) | 874.2 (58.2) | 242.6 (16.1) | 205.6 (13.7) | 179.6 (12.0) |
| Public facility, number (%) | 62,777 (64.9) | 11,414 (11.8) | 10,930 (11.3) | 11,608 (12.0) |
| Shelter outside a building, number (%) | 16 (59.3) | 6 (22.2) | 1 (3.7) | 4 (14.8) |



## 5   Results and discussion

Liquefaction phenomena were found during the Pohang earthquake in 2017. In this study, the risk levels of facilities associated with earthquake-induced liquefaction were examined for earthquake magnitudes of 5.0 and 6.5 in Kimhae City. The results are as follows.

1.  Areas with very low level of liquefaction severity for an earthquake magnitude of 5.0 cover 94% (433.5 km$^2$) of the total area in Kimhae City.  Level of liquefaction severity from high to very high are distributed in the Daedong-myun area, which consists of soft soil layers.

2.  Areas with very low and high level of liquefaction severity for an earthquake magnitude of 6.5 cover 83% (381.4 km$^2$) and 2.5% (11.5 km$^2$) of the total area, respectively. As the earthquake magnitude changes from 5.0 to 6.5, the proportions of very low and high level of liquefaction severity are 11.3% and 2.3%, respectively, whereas the proportions of low and very high level of liquefaction severity are 7.6% (35.1 km$^2$) and 6.0% (27.7 km$^2$), respectively. Moreover, the level of liquefaction severity changes from very low to low and from high to very high. Most of the areas have low level of liquefaction severity for the earthquake magnitude of 5.0, whereas some change to very high level of liquefaction severity for the earthquake magnitude of 6.5. This indicates that an $M_w = 6.5$ earthquake may result in higher risks levels for facilities associated with high level of liquefaction severity.

3.  The areas with high level of liquefaction severity for the earthquake magnitude of 5.0 cover less than 0.1% of roadways, sewage pipelines, and public facilities. In addition, 80% of facilities (except light rail transit facilities) correspond to very low level of liquefaction severity. Therefore, the liquefaction-induced risk levels for facilities are very low for the $M_w = 5.0$ earthquake. However, as the earthquake magnitude increases to 6.5, 9% of facilities (except for tunnel and railway facilities) and 30% of light rail transit facilities are distributed in high level of liquefaction severity areas, reflecting higher risk levels for these facilities.

4.  The SPT database for Kimhae City was used to estimate the CSR and LPI. Higher LPI values are found at the sedimentary layers of soils widely distributed adjacent to Nakdong river. Importantly, a magnification of ground movement occurs near the fault zone during an earthquake. Therefore, the construction of buildings in regions with high liquefaction severity should be avoided.

*Acknowledgements.* This work was supported by the 2019 INJE University research grant.

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
