# Peer review of "Brief Communication: A case study of risk assessment for"

_Natural Hazards and Earth System Sciences, 2021_

## Author Comment (AC2)

| Referee's comments | Revised contents |
|---|---|
| **Terminology**
The authors use various terms for the liquefaction potential in Kimhae City. For example, the title, heading 4.2, line 46 and elsewhere, speak about RISK; line 68 about "ground DAMAGE level"; line 52 about "estimate the HAZARDS induced by liquefaction"; line 75 about "levels of liquefaction SEVERITY". There is a need to clarify what is the evaluation about, and follow the terminology used in this discipline. | In this study, terminology is used as follows,

risk for facilities
damage for ground
hazards for ground and structures
severity for liquefaction |
| On page 5,
**Methodology**
Some aspects of the methodology are not clear, for example:

What were the criteria used for selecting the proper SPT data (Line 173) for LPI calculation; | "Since some of the important SPT data including ground depth and N-values were missing," are added in the text. |

"The SPT data of 903 locations, provided by both the geotechnical information database system of a governmental organization and construction companies, were collected to estimate the LPI values in the study area. **Since some of the important SPT data including ground depth and N-values were missing**, a reliable dataset of 274 locations was selected, and then a GIS was used to plot the locations of the selected SPT data. The locations of SPT obtained in the roads are linearly arrayed as shown in Figure 5. The SPT data recorded at the various coordinates and the kriging method were used to construct the contour lines of the LPI values."

| Referee's comments | Revised contents |
|---|---|
| On page 6,
What are the "Preliminary estimation" in the Flowchart (figure 1), and the criteria for 'yes' or 'no' decision? Similar question refers also to the criteria used for FS in the same flowchart.

There is a need to present the soil classification used for the analysis. | Equations (2) - (9) are added to support Figure 1.

Soil classification is added in the text. |

**A simplified method for estimating the FS of liquefaction was proposed by Idriss and Boulanger (2010), as follows:**

$$\text{FS} = \frac{CRR}{CSR} \tag{2}$$

**The cyclic resistance ratio (CRR) and cyclic stress ratio (CSR) represent the capacity of soil to resist liquefaction and the ratio of the shear stress relative to the effective vertical overburden stress, respectively.**

The depth-dependent shear stress reduction factor ($\gamma_d$) can be expressed as,

$$\gamma_d = e^{\left[-1.012 - 1.126 \times \sin\left(\frac{Z}{11.73} + 5.133\right) + \left(0.106 + 0.118 \times \sin\left(\frac{Z}{11.28} + 5.142\right)\right) \times M\right]} \tag{3}$$

where $z$ is a given ground depth and $M$ is an earthquake moment magnitude. CSR is expressed as,

$$CSR = 0.65 \times \left(\frac{\sigma_v}{\sigma_v'}\right) \times \left(\frac{a_{max}}{g}\right) \times \gamma_d \tag{4}$$

where $\sigma_v$ is the vertical total stress of the soil at the depth considered (kPa), $\sigma_v'$ the vertical effective stress (kPa) , $a_{max}$ the peak horizontal ground surface acceleration (g), g is the acceleration of gravity.

$$(N_1)_{60} = C_N C_E C_R C_B C_S N_m \tag{5}$$

where $C_N$ is an overburden correction factor, $C_E = ER_m/60\%$, $ER_m$ is the measured value of the delivered energy as a percentage of the theoretical free-fall hammer energy, $C_R$ is a correction factor to account for energy ratios being smaller with shorter rod lengths, $C_B$ is a correction factor for nonstandard borehole diameters, $C_S$ is a correction factor for using split spoons with room for liners but with the liners absent, and $N_m$ is the measured SPT blow count. $\Delta(N_1)_{60}$ is a function of the soil's fines content (FC) and can be expressed,

$$\Delta(N_1)_{60} = e^{\left(1.63 + \left(\frac{9.7}{FC+0.01}\right) - \left(\frac{15.7}{FC+0.01}\right)^2\right)} \tag{6}$$

$(N_1)_{60}$ can be expressed in terms of an equivalent clean-sand $(N_1)_{60CS}$, which is obtained the following expression:

$$(N_1)_{60CS} = (N_1)_{60} \times \Delta(N_1)_{60} \tag{7}$$

The magnitude scaling factor (MSF) varies with the magnitude of the earthquake($M_w$) and the following relationship:

$$MSF = 6.9e^{\left(\left(\frac{-M_w}{4}\right) - 0.058\right)} \tag{8}$$

$K_\sigma$ is the overburden correction factor can be expressed as,

$$K_\sigma = 1 - \left(C_\sigma \times \left(\frac{P_a}{\sigma_v'}\right)\right) \tag{9}$$

where,

$$C_\sigma = \frac{1}{(18.9 - 2.55 \times \sqrt{(N_1)_{60}})} \tag{10}$$

CRR is expressed in terms of $(N_1)_{60CS}$ as followings,

$$CRR = e^{\left(\frac{(N_1)_{60cs}}{14.1} + \left(\frac{(N_1)_{60cs}}{126}\right)^2 - \left(\frac{(N_1)_{60cs}}{23.6}\right)^3 + \left(\frac{(N_1)_{60cs}}{25.4}\right)^4 - 2.8\right)} \times MSF \times K_\sigma \tag{11}$$
* * *
"In this study, the LPI proposed by Iwasaki et al. (1978) was used to estimate the ground damage level induced by liquefaction. As described in Eqn. (1), the LPI is calculated based on the ground depth and characteristics of soil such as $S_c$(very dense soil and soft rock), $S_d$(stiff soil), $S_e$(soft soil), and $S_f$(soil requiring site-specific evaluation), as follows:"

| Referee's comments | Revised contents |
|---|---|
| On page 6,
Due to the poor resolution of map 4, it is not possible to identify the location of SPTs points and figure out the spread of SPT points across the city area. As far as I could see and understand, there are some areas with no data. The authors need to determine the threshold density of information relevant to the analysis, exclude no data areas from the analysis, and accordingly reexamine and modify the results presented and elaborated in Chapters 4 and 5 and in the relevant figures. | "The locations of SPT linearly arrayed inside of the dotted line may result in the deviation of contour lines of LPI as shown in Figure 4" are simplified to "The locations of SPT obtained in the roads are linearly arrayed as shown in Figure 5.".

Map 4 is magnified to be clearly viewed. |

The SPT data of 903 locations, provided by both the geotechnical information database system of a governmental organization and construction companies, were collected to estimate the LPI values in the study area. Since some of the important SPT data including ground depth and N-values were missing, a reliable dataset of 274 locations was selected, and then a GIS was used to plot the locations of the selected SPT data. **The locations of SPT obtained in the roads are linearly arrayed as shown in Figure 5.** The SPT data recorded at the various coordinates and the kriging method were used to construct the contour lines of the LPI values.
* * *
[Figure]

Figure 4. Peak ground acceleration (PGA) vs. distance from epicenter

| Referee's comments | Revised contents |
|---|---|
| On page 6,
I wonder why there is no presentation and discussion on the geology of the region. There are methods and procedures for identifying zones of required investigation for liquefaction susceptibility by geological screening, and it is thus possible to complement the investigation in region with no or scarce LPI data.

It would be useful to present the geology of the region and see whether the LPI results agree with the geology, and thus extrapolate the understandings for areas with no LPI data. | Unfortunately, the geology data are not available. In the future, if it is available, the extrapolation will be performed to validate LPI data. The characteristics of geology is indirectly reflected by SPT N value which is used to estimate liquefaction susceptibility. |
| **PGA**
The first paragraph in section '3.2 Attenuation relationship of PGA' is confusing: | It is changed to "3.2 Attenuation relationship to generate PGA" |

**"3.2 Attenuation relationship to generate PGA"**

| Referee's comments | Revised contents |
|---|---|
| On page 7, lines 12 and 13,
The text is hard to follow because there are many repetitions;
Lines 152-3 say that "Choi et al. (2005) was used in this study", while lines 159-160 state the opposite for distance shorter than 10 km, and Table 3 (line 169) base the estimation on Jo and Baag (2003).
Please rephrase and explain what were the attenuation relationships used in this study? | Choi et al. (2005) is replaced by Jo and Baag (2003).

Jo and Baag (2003) is replaced by Choi et al. (2005) in Table 3. |

"from Munson (1997), with the latter being based on ground conditions in Hawaii. As the calculated values are shown in Figure 4, as there were no available data corresponding to a distance of less than 10 km and the attenuation relationship proposed by **Jo and Baag (2003)** resulted in the overprediction of the PGAs. Eqn. (12) expresses the attenuation relationship proposed by **Choi et al. (2005)**, and Table 3 describes the parameters of the attenuation relationship for estimating PGAs."

| Referee's comments | Revised contents |
|---|---|
| The text states: (lines 114-115):
"the horizontally extended location from the centroid of Kimhae City to the closest fault is assumed to be the location of the epicenter". However, Figures 2a shows a line diagonal to the fault line rather than normal to it. The same should be applied for the 17 sub districts (Figure 2b).
Thus there is a need to correct the distances and recalculate the expected PGAs. | The text states: (**Lines 158-159**):
Figure 3a is correctly expressed how to determine the location of epicenter. Figure 3b shows the distance from the centroid of Daedong-myeon to the epicenter determined by Figure 3(a). Therefore, it is not necessary to recalculate PGA. |

(a) Distance from epicenter to the centroid of Kimhae City

[Figure]

(b) Distance from epicenter to the centroid of Daedong-myeon

Figure 3. Distance from epicenter to the centroid of Kimhae City and Daedong-myeon, respectively.

| Referee's comments | Revised contents |
|---|---|
| **Risk Level**

It appears that most of the facilities are distributed where LPI = 0. Is it an artifact due to lack of LPI data? May be there should be a minimum distance from a given facility to the nearest LPI data in order to except or reject the results.

Alternatively, are there zones with no or little LPI data but with geological conditions that favor liquefaction hazard? How would you define the hazard in such areas? | Since LPI = 0 in mountain areas mostly covered in this study area, it is not an artifact. Since LPI distributions(contours) using **kriging method** and optimal GIS mesh (cell) are generated as a polygon shape to cover all of this study area (i.e., LPI = 0 in mountain areas), it is not an artifact. The facilities are overlaid by each cell of LPI data. |
| **Results and discussion**

While defining areas with very low level of liquefaction severity in an urbanized area for an earthquake (Result 1) on the base of interpolation of LPI data but no geological screening, there should be a note that zones of significant PGA amplification, artificial landfill, leakage of water and sewage systems, etc., should be excluded and treated with care. | This study only focuses on available SPT N-values and the attenuation relationship is used to cover broad areas associated large number of N-values. The screening process with geologic data can examined for small areas or small number of N-values. However, in the future, the geological screening is very useful to validate LPI data. |
| Result 4: the authors state that "Therefore, the construction of buildings in regions with high liquefaction severity should be avoided." This is a very strict conclusion that is not fully supported in this study. Such a recommendation should be taken by an engineer after geological screening, site specific investigation, and no way for a proper soil treatment. | The statement is changed from "Therefore, the construction of buildings in regions with high liquefaction severity should be avoided." to "The methodology of the attenuation relationship used in this study doesn't cover source characteristics, propagation path, and local site conditions including presence of soft soil deposits, basin structures, and surface topography. However, it covers broad areas associated with subsequently large number of SPT N-values and may help decision-making how to develop new construction areas with respect to ground conditions resistant to earthquakes." |

   "**The methodology of the attenuation relationship used in this study doesn't cover source characteristics, propagation path, and local site conditions including presence of soft soil deposits, basin structures, and surface topography. However, it covers broad areas associated with subsequently large number of SPT N-values and may help decision-making how to develop new construction areas with respect to ground conditions resistant to earthquakes.**"

| Referee's comments | Revised contents |
|---|---|
| **Figures**
There is a need to add location map of the study area and show where Kimhae City is in South Korea, the earthquake epicenters, faults and localities mentioned in the text. | Location map of the study area (Figure 2) is added. |

Figure 2 shows Kimhae City located in Kyungsangnam-do at the southern part of South Korea.

[Figure]

Figure 2. Location of Kimhae City in South Korea

| Referee's comments | Revised contents |
|---|---|
| The maps are hard to read (I could hardly see the location of the SPTs points and other information), mainly due to low resolution and scale. Please improve resolution of the maps, text on the maps (Figures 5ab), size of legend, explain what is shown at the background of the maps, and show the limits of the urban area at the background. | Map (Figure 5) is magnified to be visible.

The resolutions and size of legend for Figure 6ab are improved.

Urban areas are described in the text but it is not appropriate to make the limits of the urban area in the figure. |

Algorithm using optimal GIS dimension developed by Jeon and O'Rourke (2005) and a kriging method has been used to determine LPI zones. Figures 6(a) and (b) show the LPI distribution and Figures 7(a) and (b) show the ratio of the covered area with respect to the range of the LPI values for $M_w = 5.0$ and 6.5 earthquakes, respectively. **Urban areas include Naeoe-dong, Hwalcheon-dong, Buwon-dong.**
* * *
[Figure]

Figure 5. Location of standard penetration test (SPT) used to estimate LPI

[Figure]

(a) Spatial distribution of LPI for $M_w$ = 5.0 earthquake

[Figure]

(b) Spatial distribution of LPI for $M_w$ = 6.5 earthquake
Figure 6. Spatial distribution of LPI for $M_w$ = 5.0 and 6.5 earthquakes, respectively

**Referee #1 (Technical)**

| Referee's comments | Revised contents |
|---|---|
| **Line 38**
Should be: "... earthquakes (Mw = 6.2, 7.1) in 2010 and 2011, respectively. | There are several sequential earthquakes between 2010 and 2011. Therefore, it is changed from "Christchurch earthquakes (Mw = 6.2 – 7.1)" to " sequential earthquakes (Mw = 6.2–7.1) in Christchurch between 2010 and 2011". |

"The soil liquefaction induced by the $M_w = 5.4$ November 15, 2017 Pohang earthquake occurred in Heunghae-eop (epicenter) was reported as a first case in Korea; however, liquefaction has occurred following various earthquakes, including the Niigata earthquake ($M_w = 7.6$) in 1964, Loma Prieta earthquake ($M_w = 6.9$) in 1989, Northridge earthquake ($M_w = 6.7$) in 1994, Tohoku earthquake ($M_w = 9.1$) in 2011, **sequential earthquakes in Christchurch earthquakes ($M_w = 6.2$-7.1) between 2010 and 2011**, Sulawesi earthquake ($M_w = 7.5$) in 2018, and Petrinja earthquake ($M_w = 6.4$) in 2020. Earthquakes resulted in substantial amounts of infrastructure damage, such as building damage induced by differential settlements, the lateral displacement of roads, and lifeline damage. The structural and"

| Referee's comments | Revised contents |
|---|---|
| **Line 175**
"inside of the dotted line" – do you mean the doted red ellipse in Figure 4? | Dotted red ellipse in Figure 5 is deleted because it is not necessary to be described. |

"The SPT data of 903 locations, provided by both the geotechnical information database system of a governmental organization and construction companies, were collected to estimate the LPI values in the study area. Since some of the important SPT data including ground depth and N-values were missing, a reliable dataset of 274 locations was selected, and then a GIS was used to plot the locations of the selected SPT data. **The locations of SPT obtained in the roads are linearly arrayed as shown in Figure 5.** The SPT data recorded at the various coordinates and the kriging method were used to construct the contour lines of the LPI values."

| Referee's comments | Revised contents |
|---|---|
| Table 4 – please round the numbers where needed. | "ea" is inserted for numbers in Table 4. |

Table 4. Facilities in Kimhae City

| Facility | Number or length |
|---|---|
| Tunnel | 15 ea |
| Bridge | 412 ea |
| Light rail transit (km) | 24.6km |
| Railway (km) | 91.3km |
| Road (km) | 1,145.3km |
| Water pipe (km) | 1,340.0km |
| Sewage pipe (km) | 1,502.0km |
| Public facility | 96,729 ea |
| Shelter outside a building | 27 ea |

| Referee's comments | Revised contents |
|---|---|
| **Line 196**
what does it mean "plat area" | It is changed from "plat area" to "relatively flat area" |

"in proportion to the total area, but are not small in proportion to the **relatively flat area**. As the earthquake magnitude"

| Referee's comments | Revised contents |
|---|---|
| **Line 238**
first sentence, seems to belong to the introduction? | The first sentence is deleted because it is not necessary to be described. |

**Referee #2 (Major)**

| Referee's comments | Revised contents |
|---|---|
| Abstract and Introduction should be significantly improved to allow the reader understanding the framework in which the topic lies, the relevance of the topic itself and the novelty of the approach proposed in the paper. It is also important to mention the advantages and also the limitations of the proposed methodology and how possible stakeholders would benefit from its application. The Title could also be more appealing. | Abstract and Introduction are revised to be concise and describe the advantages and the limitations of this study.

The title to appeal the text has been changed from "A case study of risk assessment for facilities associated with earthquake-induced liquefaction in Kimhae City, South Korea" to "GIS-based liquefaction evaluation and risk assessment of facilities in Kimhae City, South Korea" |

"**Abstract** Liquefaction causes secondary damage after earthquakes; however, liquefaction related phenomena were rarely reported until after the $M_w$ = 5.4 November 15, 2017 Pohang earthquake in Korea. Since then, a liquefaction has been an important issue in South Korea. In this study, estimations and calculations were performed using the attenuation relationship, the peak ground accelerations (PGAs) induced by $M_w$ = 5.0 and 6.5 earthquakes, and a liquefaction potential index (LPI) calculated based on groundwater level and standard penetration test results from 274 locations in Kimhae City located in the southern part of South Korea. Algorithm using optimal GIS(geographical information system) dimension and a kriging method has been used to determine LPI zones. The risk levels of facilities were evaluated based on LPI. The results indicate that a $M_w$ = 5.0 earthquake induces a small and low level of liquefaction, resulting in slight risk for facilities, but a $M_w$ = 6.5 earthquake induces a large and high level of liquefaction, resulting in a severe risk for facilities. The methodology used in this study doesn't cover source characteristics, propagation path, and local site conditions but may help decision-making how to develop new construction areas with respect to ground conditions resistant to earthquakes."
* * *
**Brief Communication: GIS-based liquefaction evaluation and risk assessment of facilities in Kimhae City, South Korea**

| Referee's comments | Revised contents |
|---|---|
| Confusion exists between the "hazard", "severity", "risk", etc. terms. Please, clarify these concepts according to the international literature (e.g. on risks associated to natural disaster) across the manuscript. | In this study, terminology is used as follows,

risk for facilities
damage for ground
hazards for ground and structures
severity for liquefaction |
| The state of the art within the Introduction needs to be significantly enriched by adding more references. To be honest, I am really surprised that none of the most relevant international references in the field of mapping the earthquake-induced liquefaction susceptibility and hazard are cited. Among others, I would like to mention the manual prepared in 1999 by the Technical Committee for Seismic Geotechnical Engineering (TC4) of the International Society for Soil Mechanics and Geotechnical Engineering, which suggests that the zoning of seismic-geotechnical hazards should be carried out according to three levels of detail and increasing refinement, which are named grade-1, grade-2 and grade-3. Recent relevant experiences in Europe should be also mentioned (see the Special Issue on the H2020 European Project LiqueFACT on Bulletin of Earthquake Engineering, 2021). In the manuscript, starting from a comprehensive and critical review of the literature, the novelty of the approach proposed by the Authors should be highlighted. Advantages and also the limitations of the proposed methodology should be mentioned in the Introduction and also in the Abstract. | Description of seismic zonation are added in the text and references.

Liquefaction hazard map was developed by Youd (1991) and the manual for zonation on seismic geotechnical hazards was proposed by the technical committee for earthquake geotechnical engineering, TC4 (1999). The architecture of the LiqueFACT (Assessment and mitigation of liquefaction potential across Europe: a holistic approach to protect structures/infrastructures for improved resilience to earthquake-induced liquefaction disasters) has been proposed by Pecker (2021). In this study, the optimal GIS mesh dimension developed by Jeon and O'Rourke (2005) has been used to establish LPI map.

1. Idriss, I.M., Boulanger, R.W.: SPT-based liquefaction triggering procedures, Department of Civil & Environmental Engineering, Report No. UCD/CGM-10-02, College of Engineering, University of California at Davis, 2010
2. Jeon, S.-S., O'Rourke, T.D.: Northridge earthquake effects on pipelines and residential buildings, Bulletin of the Seismological Society of America, 95, 295-318, 2005.
3. Pecker, A.: The H2020 European Project LiqueFACT, Bulletin of Earthquake Engineering, 19, 3803-3806, 2021.
4. Technical Committee for Earthquake Geotechnical Engineering (TC4): Manual for zonation on seismic geotechnical hazards (revised version), ISSMGE, 1-209, 1999.
5. Youd, T.L.: Mapping of earthquake-induced liquefaction for seismic zonation, 4th International Conference on Seismic Zonation, EERI, Stanford, CA., 111-147, 1991. |

**"Liquefaction hazard map was developed by Youd (1991) and the manual for zonation on seismic geotechnical hazards was proposed by the technical committee for earthquake geotechnical engineering, TC4**

**(1999). The architecture of the LiqueFACT (Assessment and mitigation of liquefaction potential across Europe: a holistic approach to protect structures/infrastructures for improved resilience to earthquake-induced liquefaction disasters) has been proposed by Pecker (2021). The optimal GIS mesh dimension has been developed by Jeon and O'Rourke (2005) to establish LPI map."**
* * *
**References**

**Idriss, I.M., Boulanger, R.W.: SPT-based liquefaction triggering procedures, Department of Civil & Environmental Engineering, Report No. UCD/CGM-10-02, College of Engineering, University of California at Davis, 2010**

**Jeon, S.-S., O'Rourke, T.D.: Northridge earthquake effects on pipelines and residential buildings, Bulletin of the Seismological Society of America, 95, 295-318, 2005.**

**Pecker, A.: The H2020 European Project LiqueFACT, Bulletin of Earthquake Engineering, 19, 3803-3806, 2021.**

**Technical Committee for Earthquake Geotechnical Engineering (TC4): Manual for zonation on seismic geotechnical hazards (revised version), ISSMGE, 1-209, 1999.**

**Youd, T.L.: Mapping of earthquake-induced liquefaction for seismic zonation, 4th International Conference on Seismic Zonation, EERI, Stanford, CA., 111-147, 1991.**

| Referee's comments | Revised contents |
|---|---|
| Earthquake-induced ground shaking is affected by: (i) source characteristics, (ii) propagation path, (iii) local site conditions, i.e. presence of soft soil deposits, basin structures, surface topography. Within the manuscript, any reference to the complexity of wave propagation is completely missing. | In the section of the results and discussions, the limitation and the advantage of this study are added as follows, "The methodology of the attenuation relationship used in this study doesn't cover source characteristics, propagation path, and local site conditions including presence of soft soil deposits, basin structures, and surface topography. However, it covers broad areas associated with subsequently large number of SPT N-values and may help decision-making how to develop new construction areas with respect to ground conditions resistant to earthquakes." |

**"The methodology of the attenuation relationship used in this study doesn't cover source characteristics, propagation path, and local site conditions including presence of soft soil deposits, basin structures, and surface topography. However, it covers broad areas associated with subsequently large number of SPT N-values and may help decision-making how to develop new construction areas with respect to ground conditions resistant to earthquakes."**

| Referee's comments | Revised contents |
|---|---|
| The paper completely lacks specific sections to illustrate the seismo-tectonic setting and the geological framework of the area under investigation. Moreover, I would expect the building of a subsoil model starting from geological information and geotechnical data. | Unfortunately, there are no geological information and geotechnical data to build subsoil model. This study focuses on LPI distributions and risk assessment of facilities based on LPI. |
| The quality of the figures especially the maps is really poor and the meaning of the map/s showing the results should be better explained within the text. | Most of figures are modified to be visible and the maps are explained by adding an additional map and additional descriptions. |
| This study completely lacks of a sensitivity analysis able to address the influence of the several assumptions carried out by the Authors on the results. Uncertainty associated to the different steps is neve mentioned. | Detailed explanations using equations are added for sensitivity analysis. |

A simplified method for estimating the FS of liquefaction was proposed by Idriss and Boulanger (2010), as follows:

$$FS = \frac{CRR}{CSR} \tag{2}$$

The cyclic resistance ratio (CRR) and cyclic stress ratio (CSR) represent the capacity of soil to resist liquefaction and the ratio of the shear stress relative to the effective vertical overburden stress, respectively.

The depth-dependent shear stress reduction factor ($\gamma_d$) can be expressed as,

$$\gamma_d = e^{\left[-1.012 - 1.126 \times \sin\left(\frac{z}{11.73} + 5.133\right) + \left(0.106 + 0.118 \times \sin\left(\frac{z}{11.28} + 5.142\right)\right) \times M\right]} \tag{3}$$

where $z$ is a given ground depth and $M$ is an earthquake moment magnitude. CSR is expressed as,

$$CSR = 0.65 \times \left(\frac{\sigma_v}{\sigma_v'}\right) \times \left(\frac{a_{max}}{g}\right) \times \gamma_d \tag{4}$$

where $\sigma_v$ is the vertical total stress of the soil at the depth considered (kPa), $\sigma_v'$ the vertical effective stress (kPa) , $a_{max}$ the peak horizontal ground surface acceleration (g), g is the acceleration of gravity.

$$(N_1)_{60} = C_N C_E C_R C_B C_S N_m \tag{5}$$

where $C_N$ is an overburden correction factor, $C_E$ = ER$_m$/60%, ER$_m$ is the measured value of the delivered energy as a percentage of the theoretical free-fall hammer energy, $C_R$ is a correction factor to account for energy ratios being smaller with shorter rod lengths, $C_B$ is a correction factor for nonstandard borehole diameters, $C_S$ is a correction factor for using split spoons with room for liners but with the liners absent, and $N_m$ is the measured SPT blow count. $\Delta(N_1)_{60}$ is a function of the soil's fines content (FC) and can be expressed,

$$\Delta(N_1)_{60} = e^{\left(1.63 + \left(\frac{9.7}{FC+0.01}\right) - \left(\frac{15.7}{FC+0.01}\right)^2\right)} \tag{6}$$

$(N_1)_{60}$ can be expressed in terms of an equivalent clean-sand $(N_1)_{60CS}$, which is obtained the following expression:

$$(N_1)_{60CS} = (N_1)_{60} \times \Delta(N_1)_{60} \tag{7}$$

The magnitude scaling factor (MSF) varies with the magnitude of the earthquake($M_w$) and the following relationship:

$$MSF = 6.9e^{\left(\left(\frac{-M_w}{4}\right) - 0.058\right)} \tag{8}$$

$K_\sigma$ is the overburden correction factor can be expressed as,

$$K_\sigma = 1 - \left(C_\sigma \times \left(\frac{P_a}{\sigma'_v}\right)\right) \tag{9}$$

where,

$$C_\sigma = \frac{1}{(18.9 - 2.55 \times \sqrt{(N_1)_{60}})} \tag{10}$$

CRR is expressed in terms of $(N_1)_{60CS}$ as followings,

$$CRR = e^{\left(\frac{(N_1)_{60cs}}{14.1} + \left(\frac{(N_1)_{60cs}}{126}\right)^2 - \left(\frac{(N_1)_{60cs}}{23.6}\right)^3 + \left(\frac{(N_1)_{60cs}}{25.4}\right)^4 - 2.8\right)} \times MSF \times K_\sigma \tag{11}$$

| Referee's comments | Revised contents |
|---|---|
| The Authors adopted only Liquefaction Potential Index (LPI, originally proposed by Iwasaki et al. 1978, 1982), but more recent lumped parameters have been proposed (e.g. Liquefaction Severity Number, LSN, etc.) and widely used in the literature. | Since SPT N-values are available in this study area, LPI is used for the analysis. Unfortunately, there are no CPT results for LSN. |
| Many sentences in the manuscript need to be substantiated by citing bibliographic references from the literature, e.g. available methods for assessing liquefaction potential from SPT, CPT, etc. I strongly recommend to adopt more then one method available for SPT data. | Since SPT N-values are available in this study area, LPI is used for the analysis. Unfortunately, there are no CPT results for LSN. |
| All the steps of the methodology are not clear in the current version of the flowchart (Figure 1), that needs to be improved, in my opinion. Please, check carefully any missing arrows and consequent step/s. | The flowchart (Figure 1) is revised to be visible and explained in details with equations as shown in previous page. |

[Figure]

Figure 1. Flowchart for estimating liquefaction potential index (LPI) (Choe and Ku, 2009)

| Referee's comments | Revised contents |
|---|---|
| Could you try to validate the map by overlapping the location of manifestations of liquefaction? | Unfortunately, there has been no liquefaction occurred in Kimhae City. Data is not available. |
| I strongly recommend to avoid to extrapolate the liquefaction hazard from such kind of maps at the locations of specific critical infrastructures. In case of specific structures/infrastructures, specific analysis is needed starting from an in-deep ground characterization of soil deposits at the site of interest. | In the future, if the analysis for specific critical infrastructures based on the ground characteristics is performed, I believe it makes much better shape of the paper. However, there are limitations to obtain the geologic data in this study area. |
| In the Conclusions, limitations and weakness points of the proposed methodology and of the presented application should be discussed in details. Concluding remarks are not fully supported in the study (see also Comment [11]). Can this methodology be applied to other areas? How? Who will be benefit from this type of maps? | Limitations, weakness points, and advantage of this study are added in the results and discussions
"The methodology of the attenuation relationship used in this study doesn't cover source characteristics, propagation path, and local site conditions including presence of soft soil deposits, basin structures, and surface topography. However, it covers broad areas associated with subsequently large number of SPT N-values and may help decision-making how to develop new construction areas with respect to ground conditions resistant to earthquakes." |

**"The methodology of the attenuation relationship used in this study doesn't cover source characteristics, propagation path, and local site conditions including presence of soft soil deposits, basin structures, and surface topography. However, it covers broad areas associated with subsequently large number of SPT N-values and may help decision-making how to develop new construction areas with respect to ground conditions resistant to earthquakes."**

**Referee #2 (Minor)**

| Referee's comments | Revised contents |
|---|---|
| The manuscript should be read carefully for English language. | English in the manuscript has been corrected. |
| Please, read carefully the paper for typing errors. | Typing errors in the manuscript are corrected. |
| Please, define the symbols, acronyms, etc. the first time you used them in the manuscript and then be consistent in the remaining text. | Modification of symbols & acronyms is carried out to define and consistently used in the manuscript. |
| With reference to the earthquake magnitude of the mentioned seismic events, please provide the source and add the references. | Since the earthquake magnitude of the seismic events is generally provided in various sources and references (i.e., Wikipedia), it is not mentioned in the text. |

| Referee's comments | Revised contents |
|---|---|
| **Line 14-26**
The abstract does not provide a concise, complete, and unambiguous summary of the work done and the results obtained. In particular, the 2016 Gyeongju earthquake mentioned in the abstract is not mentioned in the ensuing paper. Please, revise the abstract so that is going to reflect the paper contents; | Abstract is modified to be concise and to include the results.

2016 Gyeongju earthquake is deleted in the abstract. |

**"Abstract Liquefaction causes secondary damage after earthquakes; however, liquefaction related phenomena were rarely reported until after the $M_w$ = 5.4 November 15, 2017 Pohang earthquake in Korea. Since then, a liquefaction has been an important issue in South Korea. In this study, estimations and calculations were performed using the attenuation relationship, the peak ground accelerations (PGAs) induced by $M_w$ = 5.0 and 6.5 earthquakes, and a liquefaction potential index (LPI) calculated based on groundwater level and standard penetration test results from 274 locations in Kimhae City located in the southern part of South Korea. Algorithm using optimal GIS(geographical information system) dimension and a kriging method has been used to determine LPI zones. The risk levels of facilities were evaluated based on LPI. The results indicate that a $M_w$ = 5.0 earthquake induces a small and low level of liquefaction, resulting in slight risk for facilities, but a $M_w$ = 6.5 earthquake induces a large and high level of liquefaction, resulting in a severe risk for facilities. The methodology used in this study doesn't cover source characteristics, propagation path, and local site conditions but may help decision-making how to develop new construction areas with respect to ground conditions resistant to earthquakes."**

| Referee's comments | Revised contents |
|---|---|
| **Line 35**
Since the study is going to be published in an international journal, a figure introducing the study area in the geographical context of South Korea will be greatly appreciated; | Figure 2 is added to specify the relative location of the study area in South Korea. |

Figure 2 shows Kimhae City located in Kyungsangnam-do at the southern part of South Korea.

[Figure]

Figure 2. Location of Kimhae City in South Korea

| Referee's comments | Revised contents |
|---|---|
| **Lines 36-38**
In the paper, the most recent seismic events that induced liquefaction are not mentioned, e.g., 2018 Palu, Indonesia earthquake; 2020 Petrinja, Croatia earthquake; | "Sulawesi earthquake ($M_w$ = 7.5) in 2018, and Petrinja earthquake ($M_w$ = 6.4) in 2020" is added in the text. |

"The soil liquefaction induced by the $M_w$ = 5.4 November 15, 2017 Pohang earthquake occurred in Heunghae-eop (epicenter) was reported as a first case in Korea; however, liquefaction has occurred following various earthquakes, including the Niigata earthquake ($M_w$ = 7.6) in 1964, Loma Prieta earthquake ($M_w$ = 6.9) in 1989, Northridge earthquake ($M_w$ = 6.7) in 1994, Tohoku earthquake ($M_w$ = 9.1) in 2011, sequential earthquakes in Christchurch earthquakes ($M_w$ = 6.2-7.1) between 2010 and 2011, **Sulawesi earthquake ($M_w$ = 7.5) in 2018, and Petrinja earthquake ($M_w$ = 6.4) in 2020**. Earthquakes resulted in substantial amounts of infrastructure damage, such as building damage"

| Referee's comments | Revised contents |
|---|---|
| **Line 46**
The adopted LPI index has multiple drawbacks, widely known in the literature. At least a review of the most recent indexes should be included in the revised paper (e.g., (Sonmez 2003; van Ballegooy et al. 2014; Chiaradonna et al. 2020); | "Sonmez 2003; van Ballegooy et al. 2014; Chiaradonna et al. 2020" are added in the text and references. |

"Zupan, 2014). Other studies have constructed soil liquefaction hazard maps to determine land damage and/or analyze liquefaction potential (Ballegooy et al., 2012; **Ballegooy et al., 2014**; **Chiaradonna et al., 2020**; Habibullah et al., 2012; Naik et al., 2020; **Sonmez, 2003**; Ziabari et al., 2017)."
* * *
**References**

Chiaradonna, A., Lirer, S., Flora, A.: A liquefaction potential index based on pore pressure build-up, Engineering Geology, 272, 1-13, 2020.

Sonmez, H.: Modification to the liquefaction potential index and liquefaction susceptibility mapping for a liquefaction-prone area (Inegol-Turkey), Environmental Geology, 44, 862-871, 2003.

van Ballegooy, S., Malan, P., Lacrosse, V., Jacka, M.E., Cubrinovski, M., Bray, J.D., O'Rourke, T.D., Crawford, S.A., Cowan, H.: Assessment of liquefaction-induced land damage for residential Christchurch, Earthquake Spectra, 30, 31-55, 2014.

| Referee's comments | Revised contents |
|---|---|
| **Line 85-92**
 The description of the safety factor calculation is too approximate. The results are largely affected by the results (see Ramos et al. 2021 for instance), so the empirical method adopted for the calculation is not a secondary piece of information, and it needs to be specified; | The description of the safety factor calculation is added and clearly explained by using equations in the text. |

A simplified method for estimating the FS of liquefaction was proposed by Idriss and Boulanger (2010), as follows:

$$FS = \frac{CRR}{CSR} \tag{2}$$

The cyclic resistance ratio (CRR) and cyclic stress ratio (CSR) represent the capacity of soil to resist liquefaction and the ratio of the shear stress relative to the effective vertical overburden stress, respectively.

The depth-dependent shear stress reduction factor ($\gamma_d$) can be expressed as,

$$\gamma_d = e^{[-1.012 - 1.126 \times \sin\left(\frac{z}{11.73} + 5.133\right) + \left(0.106 + 0.118 \times \sin\left(\frac{z}{11.28} + 5.142\right)\right) \times M]} \tag{3}$$

where $z$ is a given ground depth and $M$ is an earthquake moment magnitude. CSR is expressed as,

$$CSR = 0.65 \times \left(\frac{\sigma_v}{\sigma_v'}\right) \times \left(\frac{a_{max}}{g}\right) \times \gamma_d \tag{4}$$

where $\sigma_v$ is the vertical total stress of the soil at the depth considered (kPa), $\sigma_v'$ the vertical effective stress (kPa) , $a_{max}$ the peak horizontal ground surface acceleration (g), g is the acceleration of gravity.

$$(N_1)_{60} = C_N C_E C_R C_B C_S N_m \tag{5}$$

where $C_N$ is an overburden correction factor, $C_E$ = ER$_m$/60%, ER$_m$ is the measured value of the delivered energy as a percentage of the theoretical free-fall hammer energy, $C_R$ is a correction factor to account for energy ratios being smaller with shorter rod lengths, $C_B$ is a correction factor for nonstandard borehole diameters, $C_S$ is a correction factor for using split spoons with room for liners but with the liners absent, and $N_m$ is the measured SPT blow count. $\Delta(N_1)_{60}$ is a function of the soil's fines content (FC) and can be expressed,

$$\Delta(N_1)_{60} = e^{(1.63 + \left(\frac{9.7}{FC+0.01}\right) - \left(\frac{15.7}{FC+0.01}\right)^2)} \tag{6}$$

$(N_1)_{60}$ can be expressed in terms of an equivalent clean-sand $(N_1)_{60CS}$, which is obtained the following expression:

$$(N_1)_{60CS} = (N_1)_{60} \times \Delta(N_1)_{60} \tag{7}$$

The magnitude scaling factor (MSF) varies with the magnitude of the earthquake($M_w$) and the following relationship:

$$MSF = 6.9e^{\left(\left(\frac{-M_w}{4}\right) - 0.058\right)} \tag{8}$$

$K_\sigma$ is the overburden correction factor can be expressed as,

$$K_\sigma = 1 - \left(C_\sigma \times \left(\frac{P_a}{\sigma'_v}\right)\right) \tag{9}$$

where,

$$C_\sigma = \frac{1}{(18.9 - 2.55 \times \sqrt{(N_1)_{60}})} \tag{10}$$

CRR is expressed in terms of $(N_1)_{60CS}$ as followings,

$$CRR = e^{\left(\frac{(N_1)_{60cs}}{14.1} + \left(\frac{(N_1)_{60cs}}{126}\right)^2 - \left(\frac{(N_1)_{60cs}}{23.6}\right)^3 + \left(\frac{(N_1)_{60cs}}{25.4}\right)^4 - 2.8\right)} \times MSF \times K_\sigma \tag{11}$$

| Referee's comments | Revised contents |
|---|---|
| The English language can be improved (e.g., line 15); | Iterated text is deleted. |

**Abstract** Liquefaction causes secondary damage after earthquakes; however, liquefaction related phenomena were rarely reported until after the $M_w = 5.4$ November 15, 2017 Pohang earthquake in Korea. Since then, a liquefaction has been an important issue in South Korea

**Referee #3 (Technical)**

| Referee's comments | Revised contents |
|---|---|
| **Line 3**
"potential" can be omitted | "Potential" is omitted. |

**Brief Communication: GIS-based liquefaction evaluation and risk assessment of facilities in Kimhae City, South Korea**

| Referee's comments | Revised contents |
|---|---|
| **Line 35**:
Pohang EQ is not introduced in the text. Please, add details (e.g., magnitude, date, epicenter) about this seismic event at the first mention in the body text; | "$M_w$ = 5.4 November 15, 2017 Pohang earthquake occurred in Heunghae-eop (epicenter)" is added in the text. |

"The soil liquefaction induced by the **$M_w$ = 5.4 November 15, 2017 Pohang earthquake occurred in Heunghae-eop (epicenter)** was reported as a first case in Korea; however, liquefaction has occurred following various earthquakes."

| Referee's comments | Revised contents |
|---|---|
| **Line 49**
FS is not defined | FS is explained by adding equations (2) - (11) in details as shown in previous page. |
| Figure 1. The flow chart is not properly discussed in the text. In particular, some parameters reported in the flow chart "$S_C$, $S_D$, $S_E$, $S_F$" remain undefined. Please, clarify this point. | $S_c$(very dense soil and soft rock), $S_d$(stiff soil), $S_e$(soft soil), and $S_f$(soil requiring site-specific evaluation) are clarified in the text. |

"liquefaction. As described in Eqn. (1), the LPI is calculated based on the ground depth and characteristics of soil such as **$S_c$(very dense soil and soft rock), $S_d$(stiff soil), $S_e$(soft soil), and $S_f$(soil requiring site-specific evaluation),** as follows:"

| Referee's comments | Revised contents |
|---|---|
| Figure 2b. The centroids of the administrative areas are not visible. Please, move the centroid layer above the shaded area of study; | Figure 3b is modified to be visible. |
| Table 2. The administrative districts are listed in Table but cannot be visualized in Figure 2. Please, rearrange the map in Figure 2a so that the name and boundary of each district can be identified; | Figure 3b is modified to identify the name and boundary of each district. |

[Figure]

(a) Distance from epicenter to the centroid of Kimhae City

[Figure]

(b) Distance from epicenter to the centroid of Daedong-myeon

Figure 3. Distance from epicenter to the centroid of Kimhae City and Daedong-myeon, respectively.

| Referee's comments | Revised contents |
|---|---|
| Figure 3. Labels in the legend cannot be read. Please, increase the figure resolution. However, the law by Choi et al. (2005) seems not reported, differently from what is said in the text. Please, revise accordingly; | Labels in the legend is modified and the figure is enlarged to be clearly visible.

Choi et al. (2005) is replaced by Jo and Baag (2003).

Jo and Baag (2003) is replaced by Choi et al. (2005) in Table 3. |

[Figure]

Figure 4. Peak ground acceleration (PGA) vs. distance from epicenter
* * *
 "from Munson (1997), with the latter being based on ground conditions in Hawaii. As the calculated values are shown in Figure 4, as there were no available data corresponding to a distance of less than 10 km and the attenuation relationship proposed by **Jo and Baag (2003)** resulted in the overprediction of the PGAs. Eqn. (12) expresses the attenuation relationship proposed by **Choi et al. (2005)**, and Table 3 describes the parameters of the attenuation relationship for estimating PGAs."
* * *
Table 3. Parameters of the attenuation relationship for estimating PGA **(Choi et al., 2003)**

| Referee's comments | Revised contents |
|---|---|
| Table 3. Numbers in table 3 are not readable. Please, revise; | Numbers in Table 3 are modified to be readable. |

Table 3. Parameters of the attenuation relationship for estimating PGA (Choi et al., 2003)

| | $\xi_0^0$ | $\xi_0^1$ | $\xi_0^2$ | $\xi_1^0$ | $\xi_1^1$ | $\xi_1^2$ |
|---|---|---|---|---|---|---|
| PGA | 0.1073829E+02 | -0.2379955E-02 | -0.2437218E+00 | 0.5909022E+00 | 0.2081359 E-03 | 0.9498274 E-01 |
| | $\xi_2^0$ | $\xi_2^1$ | $\xi_2^2$ | $\xi_3^0$ | $\xi_3^1$ | $\xi_3^2$ |
| PGA | 0.10738E+02 | -0.23799E-02 | -0.24372E+00 | 0.59090E+00 | 0.20813E-03 | 0.94982E-01 |

| Referee's comments | Revised contents |
|---|---|
| Figure 4. It is too small and the legend is unreadable. Please, enlarge the figure and increase the resolution; | Figure 5. The figure is enlarged and the legend is modified with high resolution. |

[Figure]

Figure 5. Location of standard penetration test (SPT) used to estimate LPI

| Referee's comments | Revised contents |
|---|---|
| Section 4. Line 184: The current section consists of one sentence and one table. Too short to be considered a stand-alone paragraph of the paper. Please, revise adding a detailed description of the facilities or moving the table elsewhere. | Table 4 is moved to Section 4.2 and detailed descriptions of the facilities in Table 4 are added in the text. |

**4.2 Risk assessment of facilities with respect to LPI for $M_w = 5.0$ and $M_w = 6.5$ earthquakes**

Facilities in Kimhae City are categorized as described in Table 4. Number of tunnels, bridges, public facilities, and shelter outside a building are counted and the length of light rail transit, railway, road, water pipe, and sewage pipe are specified.

Table 4. Facilities in Kimhae City

| Facility | Number or length |
|---|---|
| Tunnel | 15[ea] |
| Bridge | 412 [ea] |
| Light rail transit (km) | 24.6km |
| Railway (km) | 91.3km |
| Road (km) | 1,145.3km |
| Water pipe (km) | 1,340.0km |
| Sewage pipe (km) | 1,502.0km |
| Public facility | 96,729 [ea] |
| Shelter outside a building | 27 [ea] |

| Referee's comments | Revised contents |
|---|---|
| Figures 6 are too small and the facilities are unreadable in many cases. Please, revise. | Figures 8-10 are enlarged and modified with high resolution. |

(a)   Bridges superimposed on spatial distribution of LPI for $M_w$ = 5.0 earthquake

[Figure]

(b) Bridges superimposed on spatial distribution of LPI for $M_w$ = 6.5 earthquake

Figure 8. Bridges superimposed on spatial distribution of LPI for $M_w$ = 5.0 and 6.5 earthquakes

[Figure]

(a)  Public facilities superimposed on spatial distribution of LPI for $M_w$ = 5.0 earthquake

[Figure]

(b) Public facilities superimposed on spatial distribution of LPI for $M_w$ = 6.5 earthquake

Figure 9. Buildings superimposed on spatial distribution of LPI for $M_w$ = 5.0 and 6.5 earthquakes

[Figure]

(a)  Water pipelines superimposed on spatial distribution of LPI for $M_w = 5.0$ earthquake

[Figure]

(b) Water pipelines superimposed on spatial distribution of LPI for $M_w = 6.5$ earthquake

Figure 10. Water pipelines superimposed on spatial distribution of LPI for $M_w = 5.0$ and 6.5 earthquakes